# OpenReview forum: "Self-Jailbreaking: Language Models Can Reason Themselves Out of Safety Alignment After Benign Reasoning Training"
_ICLR.cc/2026/Conference — ICLR 2026 Poster_

### Official Review · Reviewer_9kKa · 2025-10-28

**Soundness:** 2
**Presentation:** 3
**Contribution:** 2
**Rating:** 4
**Confidence:** 4

**Summary:**

The paper proposes the concept of Self-Jailbreaking, referring to the phenomenon where certain Reasoning Language Models (RLMs) produce unsafe or inappropriate outputs after engaging in multi-step reasoning (Chain-of-Thought, CoT). Through analysis, the authors argue that this behavior originates from a process of “self-persuasion” that occurs during reasoning: as the model deliberates, it gradually compromises between safety alignment and obedience (compliance), ultimately convincing itself that the user’s intent is benign and therefore proceeding to generate harmful or policy-violating content.

The authors conduct experiments on multiple open-source Reasoning Language Models (RLMs), demonstrating the widespread nature of the self-jailbreaking phenomenon. In addition, they propose an alignment method based on lightweight fine-tuning with a small amount of safety reasoning data, which effectively mitigates such behaviors while preserving the models’ reasoning capabilities.

**Strengths:**

1.	The paper introduces the concept of Self-Jailbreaking for the first time, revealing a fundamental flaw in current Reasoning Language Models (RLMs).
2.	Experiments conducted on multiple open-source RLMs demonstrate the generality and reproducibility of this mechanism.
3.	The authors provide a theoretically grounded analysis of the underlying causes, and the interpretation based on activation-direction projection is both logical and convincing.

**Weaknesses:**

1.	The analysis of Self-Jailbreaking lacks specificity. The paper does not systematically investigate which types of questions or prompts are more susceptible to triggering this phenomenon.
2.	The proposed mitigation relies on fine-tuning the model, which limits its applicability to closed or black-box models where retraining or parameter access is not possible.
3.  The model lacks dynamic robustness. Since the proposed mechanism is based on SFT (Supervised Fine-Tuning), it remains unclear whether the model will exhibit self-jailbreaking tendencies again after long-term usage, continuous updates, or multi-turn interactions. Moreover, the paper does not discuss any decoding-time or inference-time defense mechanisms to prevent such reoccurrence.

**Questions:**

See Weaknesses

---

> ### Author Response · Authors · 2025-11-22
>
> We appreciate the reviewer’s positive feedback on our experiments and analysis, and we want to clarify on the weaknesses raised:
>
> > **W1: The paper does not systematically investigate which types of questions or prompts are more susceptible to triggering this phenomenon.**
>
> In Appendix Figure 12 (which we mentioned in line 250-251), we provided a breakdown of the prompt topics that trigger self-jailbreaking. We found that self-jailbreaking happens across different topics, suggesting that self-jailbreaking can happen across diverse malicious prompts.
>
> > **W2: The proposed mitigation relies on fine-tuning the model, which limits its applicability to closed or black-box models where retraining or parameter access is not possible.**
>
> We want to emphasize that our fine-tuning-based mitigation is intentionally designed to target the root cause of self-jailbreaking during model development, which we believe is the most effective and scalable solution. As discussed in our Discussion section (lines 472-477), our primary goal is to change current reasoning training practices in the open-source community. By demonstrating that developers can easily prevent self-jailbreaking by incorporating a small number of safety reasoning samples during training, we provide a practical solution that **prevents** unsafe models from being released in the first place.
>
> Furthermore, rather than viewing fine-tuning as a limitation, we see it as the most impactful intervention point. Model developers **already fine-tune** base models to create reasoning models (for instance, s1 models). Our mitigation simply adds minimal safety-aware data to this existing workflow. This is both practical and immediately actionable for the community.
>
> > **W3a. The model lacks dynamic robustness. Since the proposed mechanism is based on SFT (Supervised Fine-Tuning), it remains unclear whether the model will exhibit self-jailbreaking tendencies again after long-term usage, continuous updates, or multi-turn interactions.**
>
> We appreciate the reviewer’s concern about dynamic robustness, and we agree that understanding whether self-jailbreaking may re-emerge over time warrants further study. We note that existing work on mitigating emergent misalignment [1, 2] also evaluates models primarily in single-turn settings. We see our analysis as a foundation for future investigations into these more complex, long-horizon dynamics. We will clarify this limitation in our discussion section.
> .
>
> > **W3b. Moreover, the paper does not discuss any decoding-time or inference-time defense mechanisms to prevent such reoccurrence.**
>
> We want to highlight that our **activation steering experiments** (Figures 6 and 16) is a form of inference-time defense against self-jailbreaking. Specifically, we show that steering the model's activations along perceived harmfulness or compliant directions *at test-time* successfully mitigates self-jailbreaking and produces safe outputs. While additional inference-time defense mechanisms would be valuable future work, we believe that our overall contribution—identifying the phenomenon, explaining its mechanism, and demonstrating effective mitigation at both training and inference time—represents a comprehensive treatment of self-jailbreaking. We make this contribution more explicit in the Discussion section.
>
> We hope the reviewer finds our clarifications address the concerns, and we would be grateful if the reviewer could reconsider their evaluation upward in light of these responses.
>
> —
>
> [1] Wang, Miles, et al. "Persona features control emergent misalignment." arXiv preprint arXiv:2506.19823 (2025).
>
> [2] MacDiarmid et al.. “Natural emergent misalignment from reward hacking”. Anthropic asset. (2025).

---

### Official Review · Reviewer_av33 · 2025-10-31

**Soundness:** 4
**Presentation:** 3
**Contribution:** 4
**Rating:** 6
**Confidence:** 4

**Summary:**

This paper is really well-structured and highly motivative. I am really impressive by this self-jailbreaking phenomenon and get plenty of insights from this paper. The authors discover a concerning and previously uncharacterized safety failure mode "self-jailbreaking" that the reasoning model could reason itself out of safety alignment by introducing assumptions about user intent or context to justify fulfilling harmful requests or assuming that questions are only hypothetical to sidestep ethical considerations and so on. To address this, this paper shows minimal safety reasoning data during training is sufficient to ensure RLMs remain safety-aligned.

**Strengths:**

1. The investigated phenomenon is novel. Reasoning models struggle to defend against malicious queries and there are a few works[1][2][3] aiming at safety alignment improvement. However, this paper provides a new view of how and why reasoning models struggle in such settings.

2. The testing models include a wide range type, which make the conclusion convincing.

3. The self-jailbreaking rate in Figure 2 shows that this phenomenon is indeed a core reason responsible for the poor safety alignment of large reasoning models.

4. The solution is concise but effective.

[1] Improving Safety Alignment with Introspective Reasoning

[2] Safety Reasoning with Guidelines

[3] SafeChain: Safety of Language Models with Long Chain-of-Thought Reasoning Capabilities

**Weaknesses:**

1. Lack of human evaluation. Although gpt-5-2025-08-07 is a really strong model as a judge, human evaluation is still needed to reduce the  False Positive Rate and False Negative Rate.

2. Lack of baseline. I personally really like the solution of this paper that using only 50 safety reasoning samples to strongly maintain the safety and helpfulness of reasoning models. As an investigation article, I’m willing to accept that it addresses only the issues it has explored, rather than achieving state-of-the-art (SOTA) results. However, I still recommend the authors to include some safety reasoning baselines to show how this method performs against these models, though it may lag behind.

3. Lack of OOD evaluation. As this paper is investigating and solving "self-jailbreaking", it mainly focus on vanilla harmful queries. However, I am curious about whether this phenomenon exists on jailbreak harmful queries? Is the 50 safety reasoning samples training also enough for those OOD jailbreak scenes?

4. The caption in Figure 7 has squeezed the space in the following text. It is recommended to make some modifications.

I will increase my score if the above concerns are solved.

**Questions:**

See the Weakness part.

---

> ### Author Response · Authors · 2025-11-22
>
> We appreciate the reviewer’s feedback on our novelty, evaluation setup and solution, and we want to clarify on the weaknesses raised:
>
> > **W1: Lack of human evaluation. Although gpt-5-2025-08-07 is a really strong model as a judge, human evaluation is still needed to reduce the False Positive Rate and False Negative Rate.**
>
> We share the reviewer's sentiment that human evaluation is essential for ensuring reliability. However, given that it is nearly impossible to manually evaluate every CoT at the scale of our evaluation setup (hundreds of prompts with long multi-step reasoning chains for every model), we instead ask if GPT-5 is a reliable judge for self-jailbreaking.
>
> As detailed in lines 181–184, we conducted human evaluation on 250 prompts with their CoT generations, manually annotating nearly 8,300 CoT sentences. This evaluation confirmed that gpt-5 achieves over 90% precision and recall in identifying self-jailbreaking, indicating strong agreement between human and automated judgments with low false positive and false negative rates. We believe this approach provides the rigorous human verification the reviewer seeks while enabling the comprehensive coverage necessary for robust conclusions.
>
> > **W2: Lack of baseline. I personally really like the solution of this paper that using only 50 safety reasoning samples to strongly maintain the safety and helpfulness of reasoning models. As an investigation article, I’m willing to accept that it addresses only the issues it has explored, rather than achieving state-of-the-art (SOTA) results. However, I still recommend the authors to include some safety reasoning baselines to show how this method performs against these models, though it may lag behind.**
>
> Following the reviewer's suggestion, we added Appendix G3 where we benchmarked Safe-Llama-8B against SOTA safety reasoning methods on StrongReject and Wild Jailbreak datasets. Safe-Llama-8B achieves 8.1% and 24.6% ASR respectively, demonstrating **competitive performance** despite using only 50 safety reasoning samples. While SOTA methods like RECAP achieve the lowest ASR, our lightweight approach provides well-balanced robustness across both benchmarks.
>
> |                              | StrongReject | Wild Jailbreak |
> |------------------------------|--------------|----------------|
> | **SAFE-LLAMA-8B**            | 8.1          | 24.6           |
> | STAR (Wang et al., 2025c)    | 7.8          | 21.4           |
> | SafeChain (Jiang et al., 2025) | 22.3       | 47.0           |
> | DAPO (Peng et al., 2025)     | 3.2          | 27.1           |
> | RECAP (Peng et al., 2025)    | 0.3          | 11.3           |
>
>
> > **W3: Lack of OOD evaluation. As this paper is investigating and solving "self-jailbreaking", it mainly focus on vanilla harmful queries. However, I am curious about whether this phenomenon exists on jailbreak harmful queries? Is the 50 safety reasoning samples training also enough for those OOD jailbreak scenes?**
>
> We have added OOD evaluation in Appendix G.2, where we assess robustness against diverse jailbreaking attacks including adaptive attacks, past-tense attacks, prefilling attacks, and the adversarial Wild Jailbreak benchmark. Our results show that **minimal safety reasoning training provides substantial robustness improvements across these OOD attack scenarios.** For instance, SAFE-LLAMA-8B reduces ASR from 81.7% to 8.1% on StrongReject and maintains relatively low ASR (14.4-17.2%) against OOD jailbreaking attacks compared to the base model (87.2-93.8%). However, we observe higher ASR on Wild Jailbreak (24.6%), suggesting that incorporating adversarial elements beyond naive malicious prompts could further strengthen safety reasoning training.
>
> |                        | s1.1-7B | SAFE-S1.1-7B | Distilled-Llama-8B | SAFE-LLAMA-8B |
> |------------------------|---------|--------------|--------------------|--------------|
> | STRONGREJECT           | 80.2    | **4.9**          | 81.7               | **8.1**          |
> | + Adaptive Attack      | 85.4    | **14.3**         | 87.2               | **16.7**         |
> | + Past Tense Attack    | 88.6    | **16.8**         | 89.4               | **17.2**         |
> | + Prefilling Attack    | 92.1    | **14.6**         | 93.8               | **14.4**         |
> | Wild Jailbreak         | 90.7    | **19.7**         | 94.1               | **24.6**         |
>
>
> > **W4: The caption in Figure 7 has squeezed the space in the following text. It is recommended to make some modifications.**
>
> We have made changes by adding more space at the bottom of the wrapped Figure 7.
>
> We hope the reviewer finds our clarifications address the concerns, and we would be grateful if the reviewer could reconsider their evaluation upward in light of these responses.

---

> > ### Comment · Reviewer_av33 · 2025-11-22
> > **Respond to Authors**
> >
> > Thanks for this rebuttal which solves my concerns. I have updated my score to reflect my judgement of this paper.

---

> ### Author Response · Authors · 2025-11-28
>
> Thank you for the thoughtful review! We appreciate your feedback and are grateful that you raise the score.

---

### Official Review · Reviewer_TWPH · 2025-10-31

**Soundness:** 2
**Presentation:** 2
**Contribution:** 3
**Rating:** 4
**Confidence:** 3

**Summary:**

The paper reveals a new safety vulnerability in large language models, termed Self-Jailbreaking, and demonstrates that this phenomenon widely exists across various open-source reasoning language models (RLMs). Through interpretability analysis, the authors uncover its underlying mechanism and propose an effective mitigation strategy that restores the balance between safety alignment and reasoning capability.

**Strengths:**

1.It is first to systematically identify and name the phenomenon of Self-Jailbreaking, uncovering a paradoxical mechanism in which benign reasoning training unintentionally introduces safety risks — filling an important gap in current AI safety research.
2.The experiments cover multiple families of RLMs and quantitatively illustrate how internal model states evolve throughout the reasoning process.

**Weaknesses:**

1.Although the projection analysis provides intuitive evidence, it does not fully establish the causal chain between increased compliance, reduced perceived harmfulness, and self-jailbreaking; potential confounding factors may exist.
2.Projection-based interpretability has been widely used; the paper should justify its suitability and validity for analyzing self-jailbreaking specifically.
3.The study lacks quantitative experiments on different types of self-jailbreaking (e.g., hypothetical scenarios, educational rationalizations, or positive-outcome justifications).
4.The mitigation experiment (SAFE-S1.1-7B) is conducted on a single model only, without evaluating generalizability or transferability to other RLMs.

**Questions:**

1.Provide causal validation between compliance increase and harmfulness perception reduction.
2.Clarify how the proposed interpretability approach differs from existing projection-based methods.
3.Conduct proportional and quantitative analyses across different self-jailbreaking categories.
4.Reproduce the SAFE training strategy on other RLMs (e.g., Phi-4, Llama) to evaluate its generalizability.

---

> ### Author Response · Authors · 2025-11-22
>
> We appreciate the reviewer’s feedback on the importance of our work, and we want to clarify on the weaknesses raised:
>
> > **W1.Although the projection analysis provides intuitive evidence, it does not fully establish the causal chain between increased compliance, reduced perceived harmfulness, and self-jailbreaking; potential confounding factors may exist.**
>
> To address the causality concern (i.e., where self-jailbreaking sentence causes increased compliance and reduced perceived harmfulness), we perform the following experiment: during CoT generation for a particular prompt, if a CoT sentence $S_i$ is considered self-jailbreaking by our GPT-5 judge, we pause the CoT generation and regenerate $S_i$ until it is no longer a self-jailbreaking sentence. This new sentence serves as the **counterfactual**. Then, we measure the perceived harmfulness and compliance score before and after $S_i$. This controls for the potential confounding factor of model state and prompt context by keeping all preceding reasoning identical, isolating the causal effect of the self-jailbreaking sentence itself.
>
> | | Δ Compliance | Δ Perceived Harmfulness |
> |------------------------------------|--------------|-------------------------|
> | Self-jailbreaking | 3.1 ± 0.8 | −2.2 ± 0.4 |
> | Counterfactual (Non-self-jailbreaking)| −0.2 ± 0.7 | 0.1 ± 0.5 |
>
> We observe the causal effect of self-jailbreaking across 102 hand-annotated self-jailbreaking instances, where self-jailbreaking increases compliance (+3.1 ± 0.8) and decreases perceived harmfulness (−2.2 ± 0.4), while regenerated non-self-jailbreaking sentences show no significant effect, confirming self-jailbreaking as the causal driver.
>
> > **W2. Projection-based interpretability has been widely used; the paper should justify its suitability and validity for analyzing self-jailbreaking specifically.**
>
> Our projection-based analysis builds on established methods for safety analysis. [1] demonstrate that harmfulness and refusal can be represented as distinct directions in activation space, and use these directions to analyze jailbreaking in non-reasoning models. This establishes the validity of projection-based analysis for understanding safety failures, which we extend to reasoning model self-jailbreaking.
>
> We want to highlight two key novelty points of our usage of projection-based interpretability: (1) we used system prompts as it gives more controllability in safe and unsafe generations/classifications, and our Appendix G shows comparative findings with [1]; (2) we demonstrated that directions obtained from base models can analyze and steer reasoning models' behaviors during self-jailbreaking.
>
> > **W3.The study lacks quantitative experiments on different types of self-jailbreaking (e.g., hypothetical scenarios, educational rationalizations, or positive-outcome justifications).**
>
> We added Appendix F where we classify self-jailbreaking sentences for different models. Specifically, we randomly select 100 self-jailbreaking CoTs from five models and manually assign one of the six type labels given in our paper. We want to emphasize that self-jailbreaking can go beyond these observed types.
>
> We make two observation:
> - different models can exhibit varying distributions of self-jailbreaking types.
> - all the types are associated with reduced perceived harmfulness score and increased compliance, thus confirming the core findings of our work.
>
>
> > **W4.The mitigation experiment (SAFE-S1.1-7B) is conducted on a single model only, without evaluating generalizability or transferability to other RLMs.**
>
> We added to Appendix G1 our result of the mitigation experiment conducted on the Distilled Llama-3 model in, where we reproduce s1-style training on the aligned Llama-3 models. Our results are consistent with our findings on the Qwen base model.
>
> | | S1.1-7B | SAFE-S1.1-7B | Distilled-Llama-8B | SAFE-LLAMA-8B |
> |---------|---------|--------------|-------------------|---------------|
> |StrongReject (ASR) | 80.2    | **4.9**          | 81.7              | **8.1**          |
>
>
> We hope the reviewer finds our clarifications address the concerns, and we would be grateful if the reviewer could reconsider their evaluation upward in light of these responses.
>
> ---
>
>
>
> [1] Zhao, Jiachen, et al. "LLMs encode harmfulness and refusal separately." NeurIPS 2025.

---

### Official Review · Reviewer_E1ZH · 2025-11-07

**Soundness:** 3
**Presentation:** 3
**Contribution:** 4
**Rating:** 8
**Confidence:** 4

**Summary:**

They define self-jailbreaking of reasoning models as the phenomenon of RLMs reasoning their way out of safety guardrails during reasoning to assist with malicious requests, without any jailbreaking or deception attempt from the user.  They measure the occurrences of self-jailbreaking of open-weight models and how harmful their responses become after benign reasoning training on math or coding tasks. Moreover, they analyze why RLMs generate harmful outputs through self-jailbreaking and show that RLMs after benign reasoning training have increased compliance. Lastly, they show that minimal safety reasoning data can sufficiently mitigate the harmful effects of self-jailbreaking and restore safety guardrail.

**Strengths:**

- The work is the first study of self-jailbreaking
- They evaluate various models to assess how frequently self-jailbreaking appears in the reasoning models after benign math/coding reasoning training
- They try to mechanistically explain why the models show self-jailbreaking
- Moreover, they propose how to mitigate self-jailbreaking

**Weaknesses:**

I like this paper. It not only identifies an important problem but also explores a way to mitigate self-jailbreaking. Moreover, they provide a mechanistic interpretability analysis that explains why self-jailbreaking emerges in models after benign math and coding training.

However, there are some issues I notice. The paper doesn't describe details regarding the model training setup prior to the safety evaluation in Section 3.2. It is unclear what specific datasets were used, how many data points they contained, and whether a single dataset or multiple datasets were involved. In addition, the paper doesn't describe whether self-jailbreaking occurred consistently across different training datasets. Addressing these questions would be important to ensure generalization and robustness.

**Questions:**

Please see the weaknesses.

---

> ### Author Response · Authors · 2025-11-22
>
> We appreciate the reviewer's positive feedback, especially on the importance of our work and our analysis. Regarding the concern about training dataset details in Section 3.2:
>
> > **The paper doesn't describe details regarding the model training setup prior to the safety evaluation in Section 3.2. It is unclear what specific datasets were used, how many data points they contained, and whether a single dataset or multiple datasets were involved. In addition, the paper doesn't describe whether self-jailbreaking occurred consistently across different training datasets. Addressing these questions would be important to ensure generalization and robustness**
>
> We want to clarify that some evaluated models (R1-distilled, Microsoft's Phi-4, and Nvidia's Nemotron) do not release their post-training datasets due to their proprietary nature; nonetheless, we address the generalization concern through our evaluation of open-source models where training data are transparent.
>
> For instance, RStar-Coder is trained on microsoft/rStar-Coder code reasoning problems, s1 models on 1K math and puzzle reasoning problems, and UniReason-Qwen3-RL on 47k math reasoning problems of different difficulty levels. The fact that self-jailbreaking emerges across these diverse training configurations—despite differences in dataset size, domain (code vs. math vs. puzzles), and data volume (1K to 41K problems)—demonstrates that self-jailbreaking occurs **consistently** across models trained on different reasoning dataset.

---

### Author Response · Authors · 2025-11-22
**General Comments**

We appreciate the positive feedback on our work, especially on:
- **Novelty and importance**: All reviewers agree that this is the first systematic study of self-jailbreaking, which fills “an important gap in current AI safety research.”
- **Rigor**: Reviewers acknowledge our coverage of models, which demonstrates the “generality and reproducibility of this mechanism” and that ‘(Figure 2) is indeed a core reason responsible for the poor safety alignment of large reasoning models.’
- **Analysis on the self-jailbreaking mechanism**: Reviewers commended our mechanistic interpretability analysis, with one mentioning that “activation-direction projection analysis is both logical and convincing”.

Reviewers wonder if our mitigation can generalize to other model types, can defend against OOD jailbreaking attacks, and how it compares against SOTA safety reasoning training. They also ask if we can provide further causal analysis on self-jailbreaking, as well as the quantitative analysis of the types of self-jailbreaking CoT.

In response, we made the following main revisions:
1. **Section 4.2**: We added a new causal experimental results that use counterfactuals to demonstrate the causal effects of self-jailbreaking CoT on compliance and perceived harmfulness.
2. **Appendix F**: We provide a quantitative breakdown of the types of self-jailbreaking CoT, and we confirm that all of them are associated with reduced perceived harmfulness scores.
3. **Appendix G**: We provide (i) generalization of our mitigation strategy to Llama-based models; (ii) our safety reasoning against OOD jailbreaking attacks, and (iii) safety reasoning performance against SOTA methods.

---

### Meta-Review · Area_Chair_3fN2 · 2026-01-07

**Summary:**

This paper proposes a phenomenon that reasoning models will use multiple strategies to bypass their own safety and then produce harmful answers. The authors propose an analysis of such phenomena on different reasoning models and then propose a method with fine-tuning to mitigate this problem and make RLM safe again.

Strengths:

1. Provide a detailed analysis of generated reasoning traces and demonstrate that "self-jailbreaking" is one of the most important reasons leading to LRM safety failures.

2. Comprehensive experiments demonstrate the generalization of their findings and the effectiveness of their results.

Weaknesses:

In fact, reasoning models' safety problem is not a new problem, and many low-cost methods or analyses have been proposed. Also, math and reasoning fine-tuning can reduce safety, which is not a new problem. The author's claim of originality is based solely on an analysis of one type of generative problem, based on linguistic performance.


In summary, I think this paper is a comprehensive paper on LRM. Therefore, I suggest accept.

**Reviewer Concerns:**

The reviewer's concerns lie in the experiment details, baselines, evaluation, and causal arguments of their reasons. I think the authors solve most problems.

**Reviewer Scores:**

I think the reviewer will turn their attitude to positive.

---

### Decision · Program_Chairs · 2026-01-26

Accept (Poster)